# Study on Spatio-Temporal Indexing Model of Geohazard Monitoring Data Based on Data Stream Clustering Algorithm

**Jiahao Li, Weiwei Song \*, Jianglong Chen, Qunlan Wei and Jinxia Wang**

Faculty of Land Resources Engineering, Kunming University of Science and Technology, Kunming 650093, China; gahaoli@stu.kust.edu.cn (J.L.); 20222201082@kust.edu.cn (J.C.); weiqunlan@kust.edu.cn (Q.W.); 18708780866@kust.edu.cn (J.W.)
**\*** Correspondence: weiweisong@kust.edu.cn

**Abstract:** Yunnan Province, residing in the eastern segment of the Qinghai–Tibet Plateau and the western part of the Yunnan–Guizhou Plateau, faces significant challenges due to its intricate geological structures and frequent geohazards. These pose monumental risks to community safety and infrastructure. Unfortunately, conventional spatial indexing methods struggle with the enormous influx of geohazard data, exhibiting inadequacies in efficient spatio-temporal querying and failing to meet the swift response imperatives for real-time geohazard monitoring and early warning mechanisms. In response to these challenges, this study proffers a cutting-edge spatio-temporal indexing model, the BCHR-index, undergirded by data stream clustering algorithms. The operational schema of the BCHR-index model is bifurcated into two stages: real-time and offline. The real-time phase proficiently uses micro-clusters shaped by the CluStream algorithm in unison with a B+ tree to construct indices in memory, thereby satisfying the exigent response necessities for geohazard data streams. Conversely, the offline stage employs the CluStream algorithm and the Hilbert curve to manage heterogeneously distributed spatial objects. Paired with a B+ tree, this framework promotes efficient spatio-temporal querying of geohazard data. The empirical results indicate that the indexing model implemented in this study affords millisecond-level responses when faced with query requests from real-time geohazard data streams. Moreover, in aspects of spatial query efficiency and data-insertion performance, it demonstrates superior results compared to the R-tree and Hilbert-R tree models.

**Keywords:** BCHR tree; CluStream; B+ tree; Hilbert curve; Hilbert-R tree; HBase

## 1. Introduction

Yunnan Province, with its complex geological and topographical diversity, is notably prone to geologic hazards such as landslides, mudslides, and avalanches. These recurrent geohazards pose significant threats to human safety, social stability, and sustainable economic progress. As of 2020, the province had recorded 23,267 geohazards, posing risks to approximately 3,780,400 people, and causing an estimated CNY 79.673 billion in property damage. Particularly during the "14th Five-Year Plan" period, accelerated urbanization and infrastructure development have magnified the impact of anthropogenic activities on the geological environment. Moreover, climate anomalies and frequent earthquakes compound the prevailing problem of geohazards [1]. The effective monitoring, analysis, and prevention of geologic disasters necessitate an efficient spatio-temporal index. Such an index is not only crucial for real-time monitoring and early warning of geologic disasters—allowing for a quick analysis, understanding of geological event evolution, and implementation of necessary emergency measures to protect lives and property—but also aids in effectively managing and querying data on geological phenomena, changes, and trends, thereby offering valuable support for decision-making processes.

Spatial indexes (e.g., quadtrees [2], KD-tree [3], R-tree [4], grid indexes [5], etc.) are the key to realizing efficient retrieval and storage of spatial data [6–8]. In recent years, a large

number of spatial indexing techniques and methods have been proposed by domestic and foreign scholars and related researchers, with a wealth of spatial indexing techniques and methodologies emerging in recent times. Although a myriad of indexing techniques exist, the predominant dynamic spatial indexing structure in current use is the R-tree, as originally proposed by Guttman, along with its numerous variants [9–16]. These include the VoR-tree, as proposed by Mehdi Sharifzadeh, which effectively amalgamates Voronoi diagrams into the R-tree to enable efficient nearest-neighbor querying. Also worth mentioning is the LAZY R-tree, suggested by Y. Yang, which enhances the R-tree construction process with a delayed splitting method to bolster indexing efficiency. Poonam Goyal contributed to the Grid-R-tree, a merging of the R-tree with the grid, designed explicitly to cater to the querying requirements of diverse data mining algorithms, etc.

The R-tree is a variant of the B-tree-based indexing structure with a fully dynamic indexing structure. However, since the R-tree is composed based on MBR (Minimum Bounding Rectangle), the spatial objects in the index as well as the nodes at each level are represented by it, which can easily lead to rectangle overlapping, thus triggering a multiplex query situation during querying [17]; not only that, the space utilization of the R-tree's leaf nodes is also low, and the space within the nodes cannot be fully utilized [18]. For this reason, Kamel et al. [19] proposed the Hilbert-R tree, which utilizes the Hilbert curve to encode and arrange spatial objects to obtain the MBR, which reduces the overlap rate and improves the querying efficiency of the spatial data, but the shortcoming is that the performance is low when the spatial data distribution is not uniform.

Addressing the inherent limitations of R-tree's ability to handle unequally distributed data, various scholars have begun to explore the amalgamation of tree-based spatial indexing techniques with clustering methodologies. Among them, Liu et al. [20] proposed a K-means algorithm-based technique for generating a static R-tree. By leveraging the characteristics of clustering, they managed to enhance the data similarity within nodes and reduce the similarity between nodes, thus diminishing the overlap of Minimum Bounding Rectangles (MBRs). Wang et al. [21] proposed the construction of an R-tree based on the K-medoids algorithm, which compensates for the K-means algorithm's susceptibility to spatial data noise points and promotes data compactness. Jiang et al. [22], on the other hand, proposed a Gaussian Mixture Model (GMM) clustering-algorithm-based Hilbert-R tree structure. By using GMM to preprocess the spatial data, they achieved high intra-cluster data similarity and low inter-cluster similarity, ensuring that neighboring data points resided in the same leaf node while reducing the MBR overlap rate. To address the challenge of handling voluminous geological data, Yu-Hang Zhang [23] innovatively integrated the deep clustering algorithm into the construction of a Hilbert-R tree, creating an efficient data indexing structure. Huan Cheng [24], on the other hand, endeavored to expedite the storage of unevenly distributed data and the construction of rapid indexing for substantial data. To achieve this, he enhanced the K-means clustering algorithm, producing the CUK, and coupled it with the stacked long short-term memory (LSTM) model, thereby optimizing the utility of the Hilbert-R tree.

Geohazard monitoring typically deals with spatial data that are unevenly distributed. While the Hilbert-R tree, constructed through the integration of clustering algorithms, does offer expedited indexing of these data, it grapples with numerous challenges within real-time monitoring and early warning applications. Key among these is the dynamic nature of geohazard data, which necessitates real-time updating. While the Hilbert-R tree is well equipped to handle static data, it offers limited capabilities in managing the real-time updating of significant data. Furthermore, its indexing ability is largely confined to the spatial dimension, rendering it unable to satisfy the multidimensional query requirement, particularly the temporal dimension. Moreover, due to the sheer volume of data associated with geohazard monitoring, there arises a need for efficient processing and storage capabilities for large-scale data. Taking these problems into consideration, we propose an improved scheme in this paper based on the stream clustering algorithm CluStream's spatio-temporal indexing model BCHR-index, which has the following contributions:

(1)  Confronting the limitation of traditional spatial indexing, which excludes the temporal dimension, we utilize the joint B+ tree to index the temporal dimension, thereby facilitating multidimensional spatio-temporal queries;

(2)  We capitalize on the micro-clusters generated by the CluStream algorithm in our stream processing stage. In combination with the B+ tree, we construct in-memory indexes to satisfy the necessities of real-time geohazard data stream monitoring and enable a rapid response during the warning process;

(3)  We leverage the Hilbert-R tree enhanced with the CluStream data stream clustering algorithm to preprocess multidimensional spatial data. This strategy serves to minimize the areas of node MBRs and reduce their similarities, thus avoiding excessive overlap between MBRs and unnecessary multi-path retrieval during querying processes;

(4)  Employing the open-source columnar database, HBase, within the Hadoop big data processing framework, we achieve efficient storage of geohazard data.

The remainder of the paper is structured as follows: Section 2 introduces the overall model architecture and the structure of the BCHR tree. Section 3 explains the implementation of the Hilbert-R tree, based on the CluStream clustering algorithm and the multidimensional range query algorithm. In Section 4, we conduct relevant experiments on the model. Section 5 concludes the study, discussing the limitations and suggesting new directions for future work.

## 2. Model Overview

Figure 1 shows in detail the model architecture realized in this paper, i.e., the BCHR-index, which contains three main parts: the client, index layer, and storage layer. The client is mainly responsible for initiating requests and accepting responses; not directly involved in data storage and processing, it is responsible for continuously outputting real-time streaming data and sending query requests to the index layer, and the real-time streaming data will be sent to the index layer and the storage layer for processing, respectively, and before the data transmission, the client will also transform the time information of the data into Unix timestamps, and at the same time, the spatial coordinates will be transformed into a Hilbert code to facilitate the subsequent construction of the index.

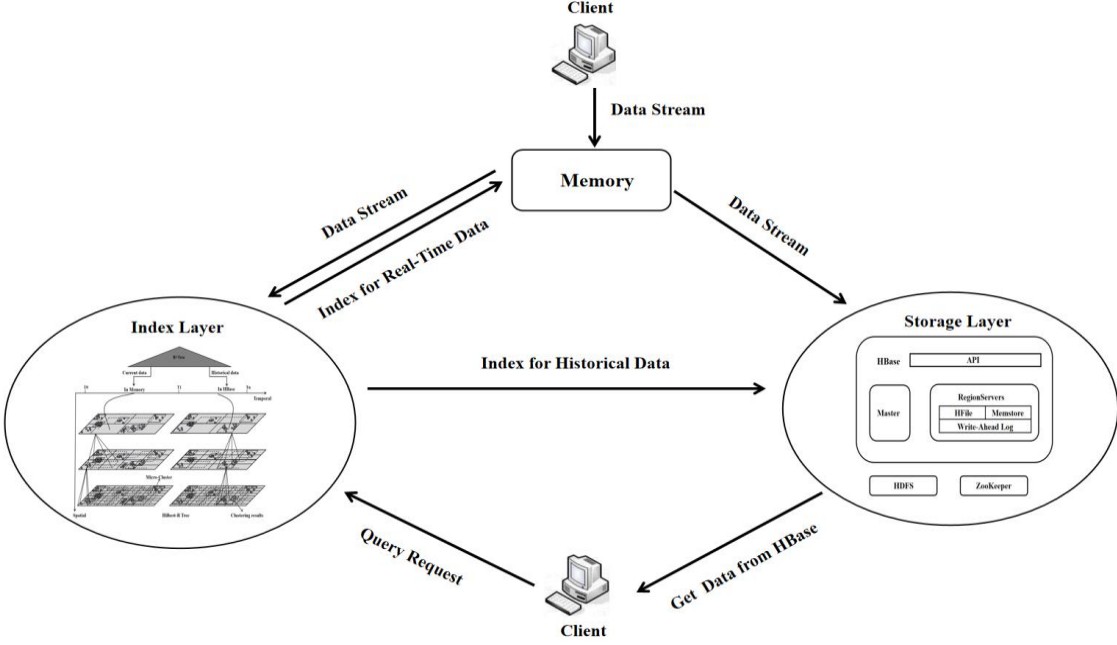

**Figure 1.** System architecture. (The Index Layer is shown in Figure 6).

Concurrently, the storage layer shoulders the responsibility of accommodating voluminous geohazard data utilizing the HBase database. We opt to store historical geohazard data in the underlying HDFS while sourcing high-incidence geohazard point data into the Block Cache. For the consistent influx of real-time streaming data from the client, the Client-side Caching function of HBase is employed to steer and inscribe the data into memory, culminating in data batch writing into HBase at fixed intervals. This methodology enhances the response speed for incoming real-time geohazard streaming data and optimizes data writing efficiency, all while diminishing the frequency of index updates. HDFS ensures that multiple copies of a single datum are dispersed across different nodes. This ensures swift data recovery through copies from other nodes even if one node fails, and this ensures business continuity and the preservation of geohazard data integrity. Meanwhile, with the growth of data volume, there is no need to make significant changes to the existing application architecture, just adding more server nodes to the HBase cluster to expand the system's storage capacity and processing capacity, which can effectively deal with a large number of geohazard monitoring data storage and access requirements, as well as read and write operations.

The indexing layer, the BCHR tree, is mainly responsible for the corresponding query operation in the face of the request sent by the client, and its work is separate from HBase, including indexing of the current data, as well as indexing of historical data in two parts. Figure 2 shows the principle and framework of the indexing layer. Four of the sub-structures play different roles, described as follows:

(1) The Hilbert-R tree serves as the principal component for facilitating spatio-temporal queries, executing spatial dimension queries based on the spatial coordinates of the objects under consideration.

(2) The CluStream algorithm processes spatial objects in the leaf nodes of a Hilbert-R tree. This technique streamlines the clustering of spatial datasets and minimizes node overlap, as well as dead space.

(3) The B+ tree is employed for indexing the time dimension of the BCHR tree, enabling the filtering of temporal information during spatio-temporal queries.

(4) The Rowkey of HBase is designed for data querying, and is a composite of the Hilbert code and Unix timestamp in this study. Utilizing the query results from both the B+ tree and the Hilbert-R tree, the Rowkey can directly pinpoint the location of data in HBase and identify the data needed to meet the query parameters.

In dealing with geohazard data, it is demonstrated that the traditional R-tree is ill equipped to support multidimensional spatio-temporal queries or respond expediently to real-time geohazard monitoring data. To address this, this paper defines a time threshold, T1. If the timestamp of incoming data is less than T1, the data are committed to memory and regarded as 'current data'; conversely, when the timestamp surpasses T1, the data are written to HBase and branded as 'historical data'. For instantaneous data, the indexing layer retrieves data cached in memory, whereas, for historical data, it queries the data stored in HBase. This strategy notably mitigates the maintenance overhead of the Hilbert-R tree. As elucidated in Figure 3, during query processing, the temporal dimension is first filtered using a B+ tree, effectively narrowing down the query range to determine the data region. Thereafter, onward queries on the spatial dimension are conducted via the Hilbert-R tree. The final querying results are a composite of the B+ tree and Hilbert-R tree queries, yielding the Rowkey of the prospective spatial object, revealing its information stored in HBase.

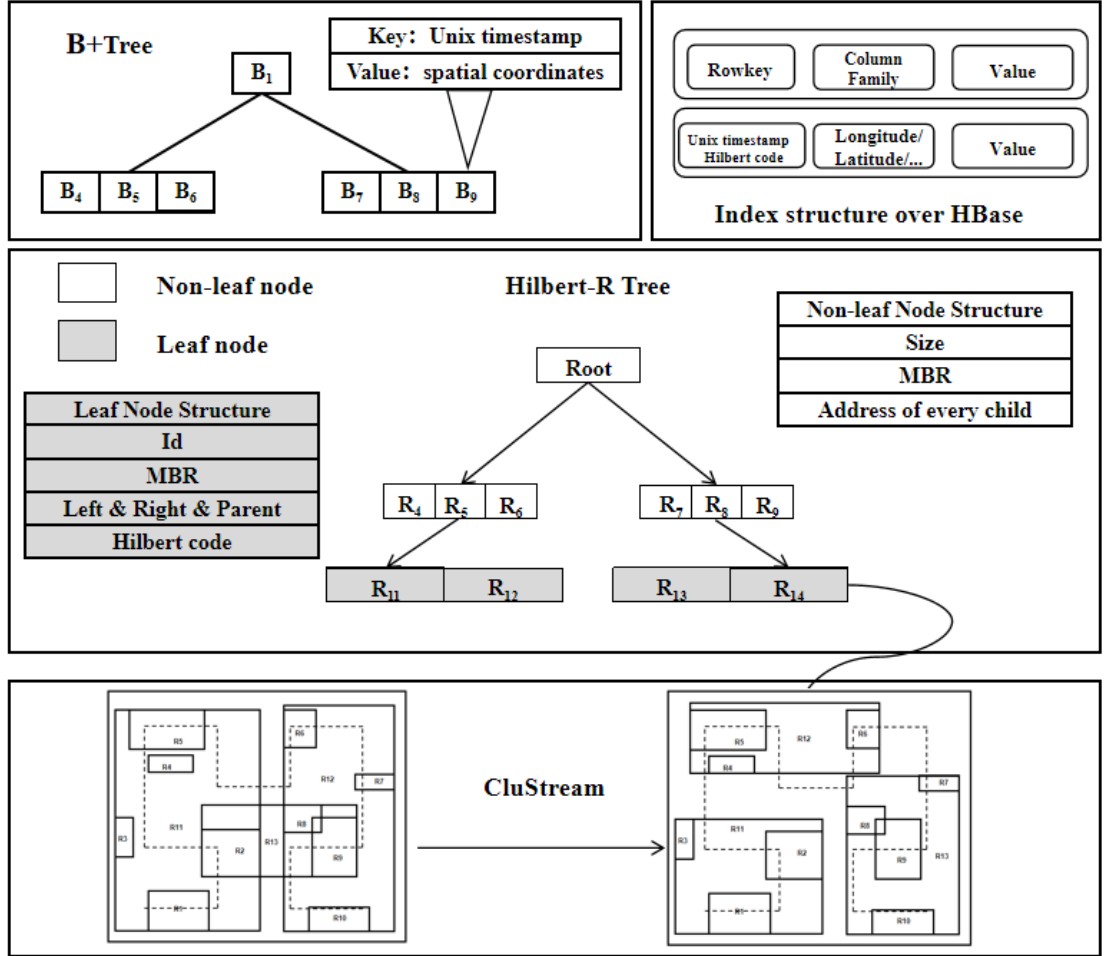

**Figure 2.** The framework of BCHR tree.

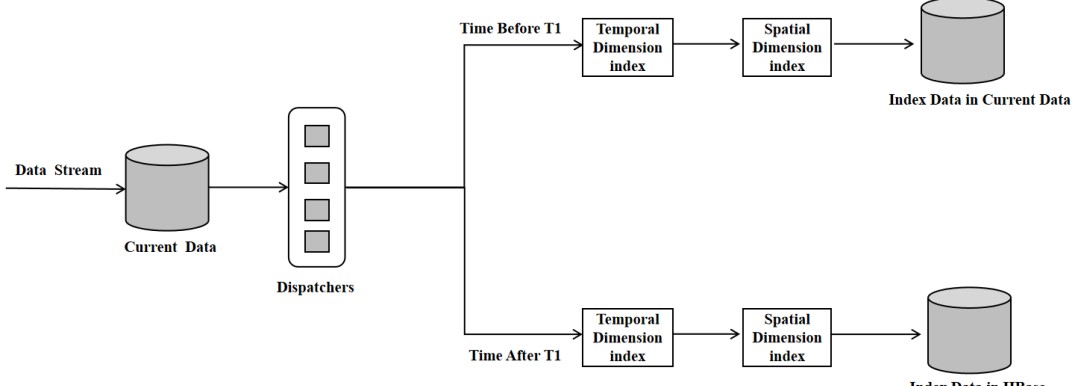

**Figure 3.** Spatial–temporal query.

## 3. Indexing Implementation Details

### 3.1. B+ Tree

Geohazard data processing frequently necessitates time-range queries. Unix timestamps, characterized as monotonically increasing integers, possess properties that perfectly complement the sorting functionality of B+ trees, facilitating an effective pairing for chronological data sorting and retrieval. Furthermore, Unix timestamps enable uncomplicated numerical comparisons for time-range queries. For the purposes of this study, we employ Unix timestamps as keys and spatial coordinates as values to construct B+ trees. The anal-

ysis utilizes 32-bit Unix timestamps, delivering millisecond-level precision. Recognizing that several geohazard events could transpire concurrently, we engineer the B+ tree as a multivariate index, wherein a single Unix timestamp can be associated with multiple spatial coordinates. This approach simplifies queries for specific temporal information. We independently construct B+ trees to index real-time data for the T0 to T1 phase. Upon storage of data into HBase, B+ trees within historic data undergo updates.

### 3.2. Hilbert-R Tree

#### 3.2.1. CluStream Algorithm

The CluStream algorithm, a stream clustering algorithm, was devised by Aggarwal et al. [25] and is renowned for its ability to process real-time data efficiently. It stands as one of the most popular baseline stream clustering algorithms [26]. The algorithm employs a two-tier processing framework, dividing the data clustering process into two stages: online micro-clustering and offline macro-clustering. The online micro-clustering stage focuses on real-time clustering of the latest data stream, while the offline macro-clustering phase implements a comprehensive clustering analysis on the complete dataset. Figure 4. illustrates the processing model of the CluStream algorithm:

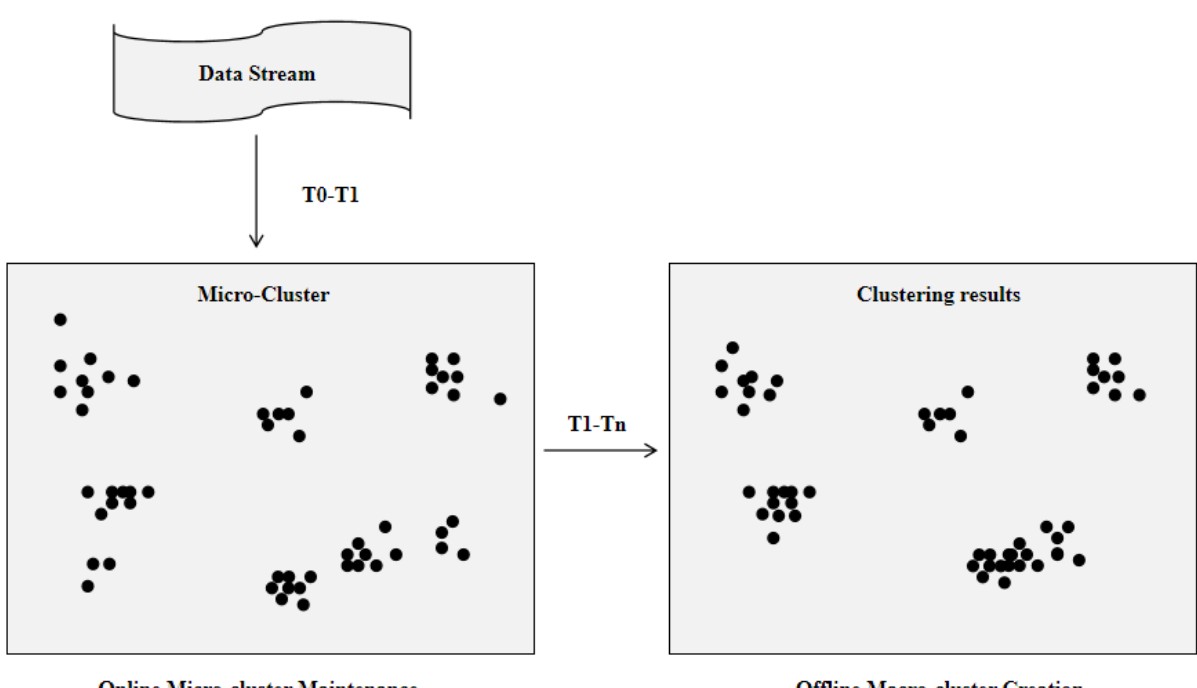

**Figure 4.** CluStream dual–layer processing model.

Online micro-clustering phase: the model first processes the selected data objects in the data stream by the k-means algorithm to generate k-initialized micro-clusters, then performs real-time clustering processing on the data stream, calculates the Euclidean distance between Xi and the center of each micro-cluster, finds the nearest micro-cluster Mq, and calculates the distance between Xi and the center boundary value of the micro-cluster Mq, and if it is within the boundary range, it will add Xi to the micro-cluster Mq. If not, a new micro-cluster is created with Xi as the center, the two closest micro-clusters are merged into one micro-cluster, and the old micro-clusters that have not been joined by new data points in the recent period are deleted.

Offline macro-clustering phase: the offline macro-clustering phase performs the final clustering of micro-clusters by the k-means algorithm based on the user input parameter t (query time) and the number of clusters K. The CluStream algorithm proposes clustering feature vectors based on the concept of a clustering feature tree, where n data points

Xi1,..., Xin forms a micro-cluster, the dimension of each data point is d, and the respective timestamps of the data points are Ti1,..., Tin. Each micro-cluster is described by a clustering feature vector (CF), defined as a 2 × d + 3 tuple with CF = ($\overline{CF2^X}$, $\overline{CF1^X}$, CF2t, CF1t, n), where $\overline{CF2^X}$ denotes the sum of squares of each data dimension within the cluster and $\overline{CF1^X}$ denotes the sum of the data of each dimension within the cluster, CF2t denotes the sum of the squares of the times of the data within each cluster, CF1t denotes the sum of the times of the data within each cluster, and n denotes the number of data points within the micro-cluster.

### 3.2.2. Hilbert-R Tree Optimized Based on the CluStream Algorithm

The Hilbert curve, a space-filling trajectory known for its exceptional locality preservation and dimension reduction properties, is used by the Hilbert-R tree to transform multidimensional data onto a two-dimensional plane [27]. By organizing Hilbert codes in a sequential manner, the Hilbert-R tree generates an R-tree, which in comparison to a traditional R-tree, reduces rectangle overlap, elevates indexing efficiency, and exudes distinct clustering characteristics [28]. Despite these advantages, the traditional Hilbert-R tree encounters challenges when dealing with unevenly distributed data within large leaf nodes, often leading to substantial overlaps and dead space.

To address the aforementioned challenges, this paper integrates the CluStream clustering algorithm into the Hilbert-R tree spatial indexing algorithm. This fusion permits a judicious organization of the spatial object clustering results and optimizes the node structure of the Hilbert-R tree [29,30]. As depicted in Figure 5, the improvements brought about by the CluStream clustering algorithm optimize spatial data division, reduce overlap and dead space, and place spatially proximate objects into neighboring leaf nodes, reducing the area of leaf and intermediate nodes and thereby enhancing space utilization.

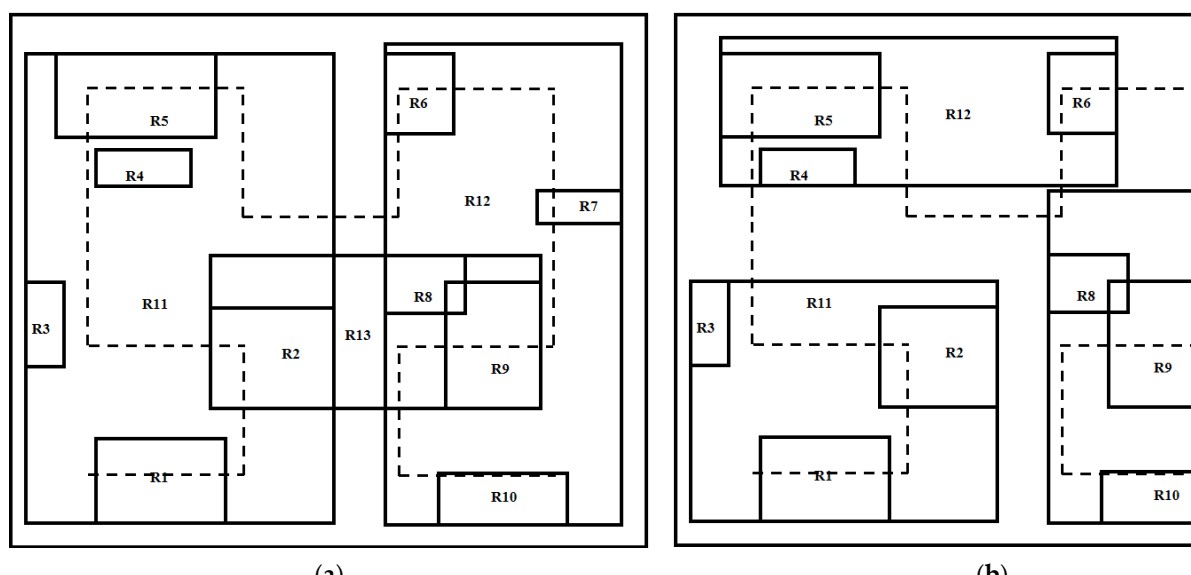

(**a**)          (**b**)

**Figure 5.** Partitioning of Hilbert-R tree leaf nodes using CluStream algorithm; (**a**) before using CluStream algorithm and (**b**) after using CluStream algorithm.

The dual-phase (online–offline) clustering processing framework of the CluStream algorithm is uniquely suited to handle the real-time stream of geohazard data [31]. Continuous geohazard stream data, generated in the time period from T0 to T1, align with the online micro-clustering phase of the CluStream algorithm. This phase equates to the time window stage of the CluStream clustering algorithm, during which the leaf nodes of the Hilbert-R tree are composed of processed micro-clusters. Conversely, data generated within the T1 to Tn time period correspond to the offline macro-clustering phase of the CluStream algorithm,

during which the leaf nodes of the Hilbert-R tree comprise macro-clustered data. Such a design ensures a swift response to real-time geohazard data streams. Combined with the B+ tree mentioned earlier, the structure of a complete BCHR tree is illustrated in Figure 6.

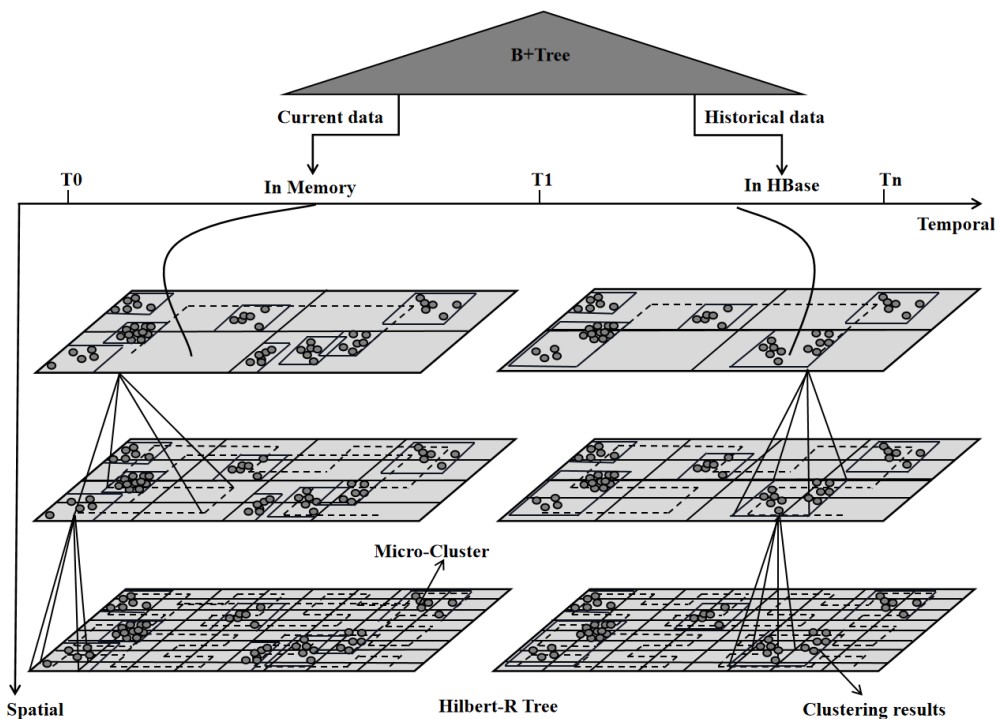

**Figure 6.** BCHR tree.

Compared with the traditional spatio-temporal index, the BCHR tree adopts a two-layer processing framework, which is designed to consider the characteristics of a large volume, uneven spatial distribution, and real-time updating of geohazard monitoring data, and is constructed by combining the B+ tree and the Hilbert-R tree based on the CluStream algorithm, which has a strong ability to process large-scale geospatial data in real time. During a spatio-temporal query, the B+ tree is employed to index the temporal dimension—essentially filtrating the time data. With the advantage of sequential access characteristics, it swiftly pinpoints the precise time phase of the query subject before proceeding with the corresponding spatial query. When the query is confined to the spatial dimension solely, there is no requirement for the application of the B+ tree for temporal dimension filtration, thereby further boosting the query efficiency. This conflation can be suitably tailored to match diverse business stipulations.

During the two stages of the query process, the BCHR tree utilizes the CluStream algorithm in the real-time phase to build indices in the memory based on the micro-clusters generated in the online stage, effectively reducing both the time required for the data query and process, and the system's storage demand. Given the relatively small data volume of real-time streaming data in the real-time processing stage and the rapid index creation speed, the BCHR tree is noted for greatly enhancing the efficiency of real-time monitoring of geological hazards, illustrating its robust real-time performance. In the offline stage, a substantial quantity of geohazard monitoring data is stored in HBase, which can be directly accessed through the BCHR tree query results, bypassing the need for additional time expenditure. This proposes an efficient storage mechanism for large-scale geohazard monitoring data and facilitates prompt retrieval and real-time updating capabilities.

In the realm of spatial data handling, the BCHR tree employs the CluStream algorithm and Hilbert curve for spatial object treatment, clustering geographically proximate data within a specific time range. The data are symbolized by the cluster centers, act-

ing as the index tree nodes, thereby optimizing the query efficiency. This approach safeguards the continuity and integrity of the spatial distribution of geological hazards, elevating the spatial query efficiency during geological hazard monitoring. When real-time data mature into historical data, specifically when data from the T0 to T1 phase evolve into data from the T1 to Tn phase, an update of the index tree ensues. There exists no necessity for frequent amalgamation, deletion, and other operations pertaining to BCHR tree nodes, which reduces computational burden and amplifies the system's efficiency in handling real-time data. These highly advantageous features of the BCHR tree greatly streamline the process of geological hazard monitoring and provide innate benefits for early warning systems.

### 3.2.3. Algorithm for Generating the BCHR Tree

The main part of the BCHR tree is the construction of the Hilbert-R tree in conjunction with the CluStream algorithm. Algorithm 1 provides the methodology for the generation of the BCHR tree. The algorithm requires four input parameters: the dataset S, the time period T, the number of clusters K, and the maximum number of data objects that each node can hold M. In the initial phase, an empty tree, T, is first created, and then each object O in the dataset S is processed to find its Minimum Bounding Rectangle (MBR) and compute its center. Next, the CluStream algorithm is applied to generate K clusters based on these centers. Subsequently, using the MBR centers of the objects, the Hilbert values for each time period (T0 to T1 for micro-clusters and T1 to Tn for macro-clusters) were computed and the generated clusters were sorted in ascending order of Hilbert values. In the sorted set of clusters, its Hilbert value is calculated for each cluster center. If the number of spatial objects within cluster C does not exceed M, a leaf node is created through all spatial objects in the cluster. Conversely, if it exceeds M, all spatial objects are sorted according to their Hilbert values, and groups containing M spatial objects are created (the last group may contain less than M objects). A leaf node is then created for each object group. These leaf nodes are further sorted based on their Hilbert values and subsequently inserted into the Hilbert-R tree. Ultimately, the Hilbert-R tree is constructed from the bottom to top using these leaf nodes.

---

**Algorithm 1**: GenerateBCHRTree

---

**Require**: Dataset S, Time segmentation T, Number of clusters K, Maximum capacity of a node M
**Ensure**: Hilbert-R Tree
Initialize an empty tree T
**for** each object o in S **do**
      Compute its MBR and the center of its MBR
      Use CluStream algorithm to generate K clusters based on these centers
      Calculate all objects' Hilbert values from their MBR centers in time frames ($T_0$ to $T_1$ for micro-clusters and $T_1$ to $T_n$ for macro clusters)
**end for**
**for** each cluster c in Clusters **do**
      Calculate the Hilbert value of the cluster center
      **if** number of space objects in c $\leq$ M **then**
      Create a leaf node by all space objects in
    **else**
      Sort all the space objects according to their Hilbert values in ascending order
      Create groups containing M space objects (the last group may have objects < M)
      Create a leaf node for each grouped space objects
    **end if**
      Sort the leaf nodes to be inserted into the Hilbert R-tree based on Hilbert values
**end for**
From bottom-up, construct the Hilbert R-tree using the sorted leaf nodes
**return T**

---

3.2.4. Spatio-Temporal Range Query Algorithm for BCHR Tree

Spatial range queries and spatio-temporal range queries are frequently utilized querying methodologies. These methods play a pivotal role in facilitating quick responses to geohazards and proficient management of geohazard activity. The following section will introduce these two query algorithms in detail:

(1)    Spatial scope query

Algorithm 2 delineates the processing steps of the spatial extent query algorithm within the BCHR tree. The primary objective is to pinpoint rectangles that intersect in a delineated bounding box. The algorithm proceeds by iterating through each entry, denoted as 'e', encompassed within the node 'n'. For each 'e', the algorithm calculates the corresponding Minimum Bounding Rectangle (MBR) and evaluates whether it intersects with the bounding box 'bb'. If 'n' is a leaf node, the algorithm incorporates the intersecting rectangle into the output 'results'. On the other hand, if 'n' is an internal node, the intersecting rectangle signifies a subtree. The algorithm continues the IntersectSearch process, recursively over the 'e.node' and the bounding box 'bb', amalgamating any discovered intersecting rectangles into 'results'. Ultimately, the algorithm returns the result set 'results', encompassing all intersecting rectangles found within the subtree of the node 'n' and bounding box 'bb'.

---

**Algorithm 2**: IntersectSearch

---

**Result**: results
**Input**: n (Node), bb (Bounding Box)
**Output**: results
results = {}; // An empty list to hold Rectangles found within 'bb'.
**for** each entry e within the node n do
    Get the Minimum Bounding Rectangle (MBR) for the entry e.
    **if** bb intersects with the MBR of the entry e then
      **if** n is a leaf node then
        results = results ∪ e;
      **else**
        results = results ∪ IntersectSearch(e.node, bb);
    **end if**
  **end if**
**end for**
**return** results;

---

(2)    Real-time temporal and spatial range queries

Traditional R-trees and their variants fall short of supporting real-time spatio-temporal range queries. To overcome this limitation, we design a spatio-temporal indexing model that merges the B+ tree and CluStream algorithm, thereby enabling enhancements in this regard [32]. As elucidated in Algorithm 3, upon receiving a query request from a client, the model initially retrieves the micro-clusters, all the data consolidated up to the current time within a given time window, from the CluStream algorithm. Subsequently, the model constructs B+ trees as well as the Hilbert-R tree based on the fetched micro-cluster files. The B+ tree is employed to perform temporal range queries while the Hilbert-R tree is utilized for spatial range queries. The separately retrieved results are then compared, and the outcomes meeting both the temporal and spatial query conditions are returned.

---

**Algorithm 3**: RealTimeSpatialTemporalQuery

---

**Result**: finalResults
**Input**: bb (Bounding Box), t1, t2
Initialize microClusters ← Read from CluStream algorithm
Initialize HilbertRT ← Constructed by microClusters
Initialize BplusTree ← Constructed by (time, microCluster) pairs from microClusters
Initialize timeResults ← searchRange(BplusTree, t1, t)
Initialize spaceResults ← searchIntersect(HilbertRT, bb)
Initialize finalResults ← ∅
**for** eachresultintimeResults **do**
**If** result ∈ spaceResults **then**
  finalResults ← finalResults ∪ result
**end**
    **end**
**return** finalResults

---

## 4. Experiments

### 4.1. Experimental Design

To evaluate the real-time query performance of the proposed spatio-temporal indexing model, the BCHR tree, and its optimization impact on inhomogeneous data for the Hilbert-R tree, we carried out a comparison study with BCHR tree, Hilbert-R tree, Elasticsearch, and R-tree. The specific experimental environment is detailed in Table 1.

**Table 1.** Hardware Configurations and Software Environment of the Experimental Platform.

| Category | Configuration |
|---|---|
| Number of VM | 3 |
| Processor | Intel Core (4 cores) |
| RAM | 8 GB |
| Hard Drive | 50 GB |
| Operation System | Centos 7.5 |
| Hadoop Version | 3.1.3 |
| Zookeeper Version | 3.5.7 |
| HBase Version | 2.4.11 |
| JDK Version | 1.8.0_212 |
| Elasticsearch Version | 7.8.0 |

The experimental data used in this paper were generated by simulation based on the current status of geologic hazards in Yunnan Province in the "14th Five-Year Plan for the Prevention and Control of Geologic Hazards in Yunnan Province". There are 23,267 geohazard sites, including 17,450 landslides, 2237 avalanches, 3118 mudslides, 331 ground collapses, 21 ground subsidence sites, and 110 ground cracks (Reviewer 1 and question 4 (a) are shown in Table 2.

**Table 2.** Description of the experimental dataset.

| Disaster Time | Timestamp | X Coordinate | Y Coordinate | Disaster Type |
|---|---|---|---|---|
| 1 January 2000 0:20 | 946657230 | 97295417 | 26758213 | Landslide |
| 15 February 2017 2:04 | 1487095493 | 98463226 | 24882692 | Mudslide |
| 25 February 2018 10:24 | 1519525491 | 103810338 | 22889044 | Collapse |
| 26 April 2019 10:13 | 1556244796 | 99869433 | 21967614 | Land Subsidence |
| 29 May 2022 13:43 | 1653803032 | 104122763 | 24934143 | Earth Cracker |

Each record of the data possesses five attributes, namely the Disaster Time, timestamp, longitude, latitude, and disaster type. The Disaster Time spans from the year 2000 to 2023 with a precision of seconds. The Timestamp represents the corresponding Unix timestamp, and during this period, it is assumed that multiple geological disasters can occur at any

given time. The longitude and latitude are simulated based on the geographical range of Yunnan Province: the longitude ranges from 97°31′ E to 106°11′ E, and latitude ranges from 21°8′ N to 29°15′ N. These latitude and longitude data, after being converted to decimals, have been multiplied by 1,000,000. Such a process is intended to translate floating-point numbers into integers and attains meter-level precision in the simulated data. The disaster type incorporates a landslide, collapse, debris flow, ground collapse, subsidence, and ground fissure. Each disaster point is randomly assigned with one of the six disaster types. The geohazard prone points and their surrounding geographic coordinates exhibit a higher probability of geological disaster occurrence. The specific algorithm is shown in Algorithm 4:

---

**Algorithm 4**: GenerateGeologicalDisasterData

---

**Result**: geodisaster_dataset
**Input**: Defined_hazards, total_num_of_disasters, timerange, coordinaterange, hazard_prone_locations
Initialize geodisaster dataset to empty list
**for** counter < total_num_of_disasters **do**
    Randomly select a hazard from Defined_hazards
    Generate random Disaster Time within timerange and convert to Unix Timestamp
    Generate random Longitude and Latitude within coordinaterange
    Convert Longitude and Latitude to decimal
    Multiply by 1,000,000 to avoid floating point and ensure meter level accuracy
    **If** current location is in hazard_prone_locations or its surrounding coordinates **then**
        Increase the probability of this Disaster Type
    **end If**
    Add record (Disaster Type, Disaster Time, Timestamp, Longitude, Latitude) to geodisaster_dataset
Increment counter
**end for**
**return** geodisaster_dataset

---

When it comes to the determination of K in the CluStream clustering algorithm, in our research, it has been decided that the number of clusters, K, should be set at 1% of the size of the dataset. This decision stems from several considerations. First, by designating K as a percentage of the dataset size, we can assure that the number of clusters scales appropriately with the increase in the dataset size, thereby accommodating better to different scales of data. Second, selecting 1% as the ratio provides a balance between the precision of clustering and the efficiency of computation. Indeed, a larger K could yield a more granular clustering, but at the expense of an exponential increase in computational complexity. Conversely, a smaller K might diminish computational complexity, but could possibly oversimplify important data characteristics. Consequently, we choose 1% as a setting that can both guarantee a certain level of clustering precision and maintain computational efficiency.

### 4.2. Comparative Experiments and Analysis of Results

4.2.1. Performance of Real-Time Spatio-Temporal Query

Real-time spatio-temporal range queries play a crucial role in the prevention and timely warning of geological disasters. This section utilizes Algorithm 4 to simulate real-time geological data streams of 5000, 10,000, 15,000, 20,000, and 25,000 records to assess the real-time query performance of the BCHR tree. The time range spans from 0:00:00 on 1 October 2023 to 23:59:59 on 1 October 2023. The spatial coordinates in terms of longitude and latitude range respectively from 98°53′ E to 104°33′ E, and 22°33′ N to 28°48′ N. In this experiment, "data size" signifies the proportion of spatio-temporal range queries in the overall data. For instance, a "20% data size" implies that the spatio-temporal range query condition encompasses 20% of the total temporal and spatial range. Conversely, the "Number of tasks" denotes the quantity of data required for constructing the index.

Elasticsearch is an open-source, distributed real-time search and analysis engine, boasting exceptional real-time search capabilities and scalability [33]. However, as illustrated in Figure 7a, when dealing with 10,000 geological disaster monitoring simulation data points, the BCHR tree outperforms Elasticsearch in real-time query performance under

different spatio-temporal range queries. This superior performance can be attributed to the BCHR tree's dual-processing framework. Under real-time query demands, the BCHR tree constructs the index tree by acquiring the clusters within the current CluStream algorithm's time window. Concurrently, the CluStream algorithm imparts similarity amongst the spatial objects in each leaf node of the BCHR tree. This feature lends a distinct advantage, especially when dealing with non-uniformly distributed data typified by geological disasters. In the process of real-time querying, the BCHR tree actually spends very little time in the querying process, and it mainly spends time on the construction of the index tree, which can be seen in Figure 7b. Figure 7c further demonstrates the BCHR tree index construction performance under different sizes of data flow scenarios. When faced with extremely scattered and uneven geohazard data, the BCHR tree clearly outperforms the Hilbert-R tree. The main reason for this is that the Hilbert-R tree has more unfilled leaf nodes when dealing with this scenario, which increases the time spent on building the tree.

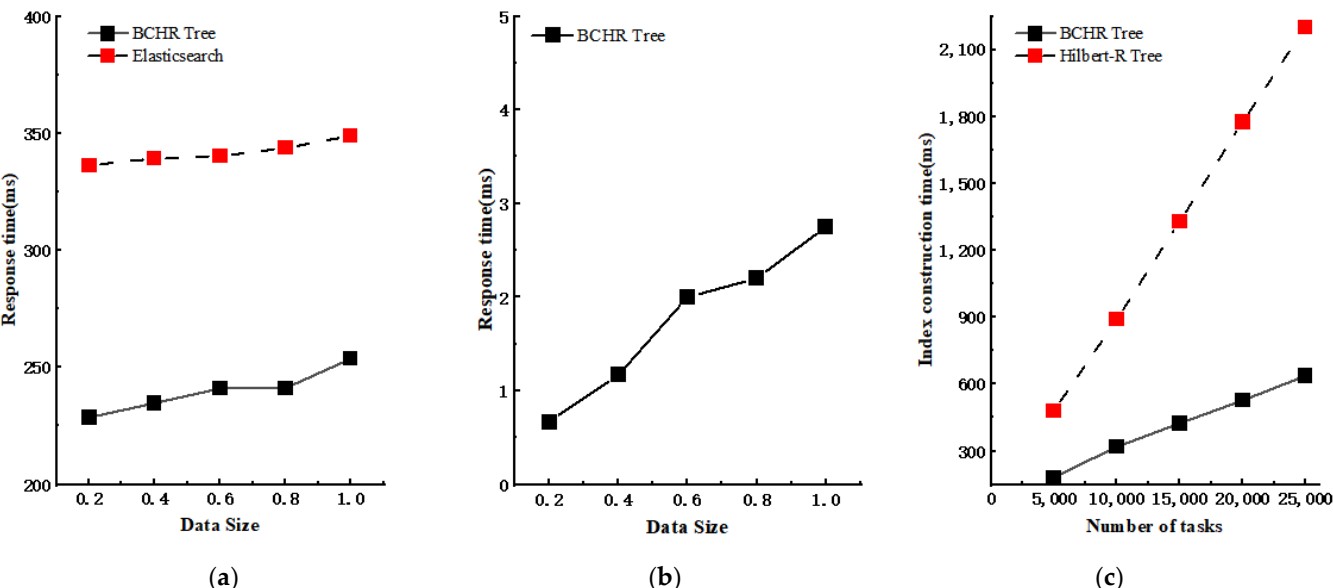

**(a)**    **(b)**    **(c)**

**Figure 7.** The performance of the BCHR tree in real-time spatio-temporal range queries: (**a**) Time spent on spatio-temporal queries for BCHR tree and Elasticsearch built on 10,000 data points with different data sizes. (**b**) Time spent on spatio-temporal queries for BCHR tree built on 10,000 data points with different spatio-temporal scopes. (**c**) Time spent on building indexes for BCHR tree and Hilbert-R tree with different data sizes.

The experiment's findings affirm that the BCHR tree can stably handle different volumes of geohazard data, keeping the response time at the millisecond level. This performance demonstrates that the BCHR tree is well equipped for effective real-time spatial–temporal queries in geological flow data.

### 4.2.2. Performance of Spatial Range Queries

Geohazards vary significantly in their frequency, intensity, and type across temporal and spatial dimensions, typifying an instance of inhomogeneous data. This study segment primarily investigates the optimization efficacy of the BCHR tree compared to the Hilbert-R tree and R-tree when dealing with static inhomogeneous data. The experimental data represent simulated geohazard scenarios throughout Yunnan Province from the year 2000 to 2023. It consists of 1 million items, and the spatial query range is set to 20%, 40%, 60%, 80%, and 100% of the total data size, respectively.

Figure 8 demonstrates an evident optimization in the query performance of the BCHR tree when compared to the Hilbert-R tree and the R-tree, while handling millions of geohazard data items. It is discernible that as the spatial data query range progressively

increases, the performance of the proposed BCHR tree maintains a level of stability against the query times of the Hilbert-R tree and R-tree.

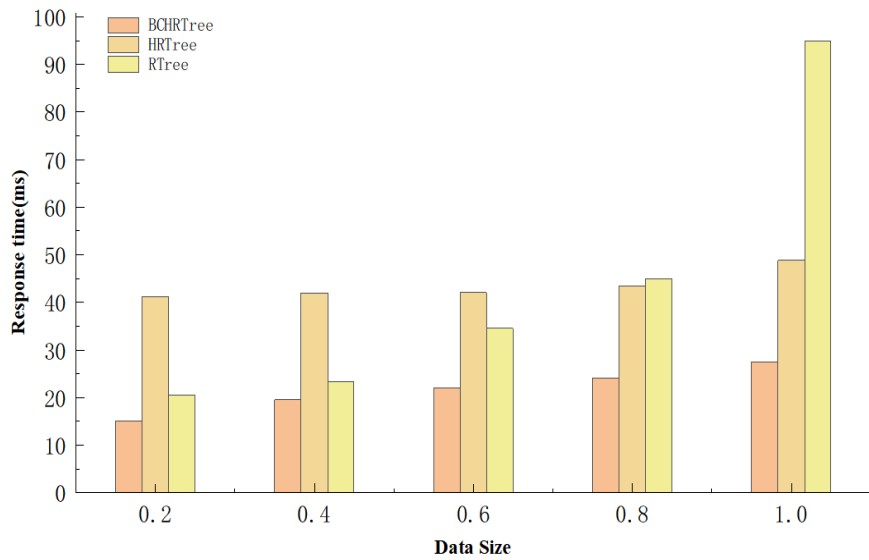

**Figure 8.** The performance of the BCHR tree in Spatial Range Query.

4.2.3. Performance of Index Insertion

In the face of continuously generated geological data, the maintenance and updating of the index is criminally essential. This section evaluates the insertion performance of the BCHR tree, where we initially constructed the BCHR tree and the Hilbert-R tree. Both trees were created based on 1 million simulated geological disaster data points generated by Algorithm 4, with an initial data volume of 1 million. The temporal period ranges from 00:00:00 on 1 October 2023 to 23:59:59 on the same day. The spatial coordinates range between East longitude 98°53′ to 104°33′ and North latitude 22°33′ to 28°48′. On this basis, we evaluated the response times of the BCHR tree and Hilbert-R tree when inserting 100,000, 200,000, 300,000, 400,000, and 500,000 geological disaster data points, respectively. We also accounted for the number of node split occurrences during the insertion period. The insertion data were generated using the same method, and Figure 9 displays the insertion performance of the BCHR tree and Hilbert-R tree.

From the experimental results, we note that the BCHR tree demonstrates a distinct improvement in insertion performance over the Hilbert-R tree. The primary reason lies in the CluStream algorithm's potential to effectuate efficient clustering of spatial objects. Spatial objects situated in close proximity are grouped into the same cluster, with contiguous leaf nodes being constructed. During the insertion process, it reduces the number of leaf nodes necessitating comparison by the insertion algorithm. Furthermore, per the BCHR tree's construction algorithm, all spatial objects within the same cluster are rendered into the same leaf node. If this node reaches capacity, the data are distributed to neighboring leaf nodes. This results in a large number of leaf nodes remaining under capacity. Also, spatial objects dispersed across space forming clusters inevitably lead to under-capacity leaf nodes. This consequently minimizes the number of node splits instigated by inserting objects, thereby enhancing the BCHR tree's insertion performance.

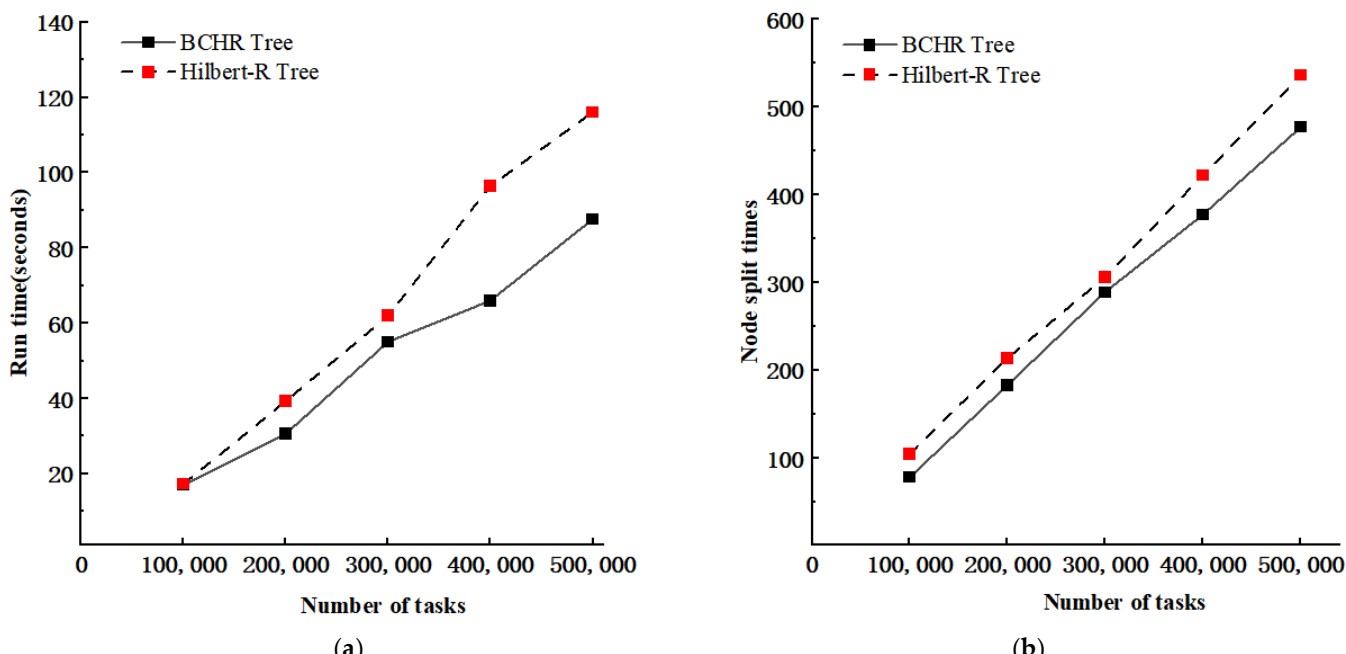

**Figure 9.** The performance of insertion operations in BCHR tree and Hilbert-R tree: (**a**) Time taken for insertions. (**b**) Node split occurrences during insertion.

## 5. Conclusions and Outlook

Confronted with the real-time generation of massive geological disaster data, there is an imperative need for an efficient real-time stream data processing framework to satisfy the rapid response demand of real-time monitoring and early warning of geological disasters. As one of the most widely used spatial index structures, the R-tree exhibits commendable performance in dealing with static data, yet it struggles with handling streaming data and does not flexibly cater to temporal indexing needs. Consequently, this study proposes a spatio-temporal index model based on a data stream clustering algorithm, the BCHR-index, to meet the requirement for multidimensional spatio-temporal queries of geological disaster data. The BCHR-index model harnesses the properties of the stream clustering algorithm CluStream and employs a real-time/offline two-tier processing framework paired with a B+ tree to construct the BCHR tree, partitioning data into real-time and offline stages. Thanks to the small data volume of the real-time data stream, the CluStream-method-generated micro-clusters can construct indices in real-time, enabling nearly instantaneous responses to geological streaming data. The offline phase builds a Hilbert-R tree using spatial data processed with the clustering algorithm, utilizing the cluster centers as leaf nodes. This maintains the continuity and integrity of the spatial distribution of geological disasters, enhancing the spatial query efficiency during the monitoring process. Even when dealing with unevenly distributed geological disaster data, the model boasts millisecond-level response times. Taking into account the sheer volume of geological disaster monitoring data, the model leverages HBase for storing such data, ensuring a certain degree of fault tolerance and scalability. However, improvements can still be made to the model. In future works, (1) we plan to further enhance the real-time indexing. Despite the millisecond-level responses of the index presented in this study, each real-time query requires the reconstruction of the index, thus adding a temporal overhead to a certain extent. (2) We will explore the best way to select the K value; the CluStream algorithm uses the K-mean method to generate clusters, and when using the CluStream algorithm to process the leaf nodes of the BCHR tree, this paper selects the K value of 1% of the number of datasets, but different types of geohazards and occurrence areas; the most suitable K value is different, so the K value selection needs to be more flexible and variable. (3) Finally, future investigations

will look to enhance the robustness of our system, ensuring rapid and accurate responses in emergencies to protect people's lives and properties.

**Author Contributions:** Conceptualization, Jiahao Li; Software, Weiwei Song, Jianglong Chen, Qunlan Wei, and Jinxia Wang; Validation, Jiahao Li and Weiwei Song; Writing—original draft, Jiahao Li; Writing—review and editing, Jiahao Li and Weiwei Song. All authors have read and agreed to the published version of the manuscript.

**Funding:** This research was funded by the Yunnan Province Key Research and Development Program (No. 202202AD080010).

**Data Availability Statement:** The data that support the findings of this study are available on request from the corresponding author.

**Acknowledgments:** The authors would like to express their sincere appreciation to all those who have offered valuable recommendations and comments that significantly improved the quality of this manuscript.

**Conflicts of Interest:** The authors declare no conflicts of interest.

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
