# Peer review of "Study on Spatio-Temporal Indexing Model of Geohazard Monitoring Data Based on Data Stream Clustering Algorithm"

_ijgi, doi:10.3390/ijgi13030093_

Round 1
Reviewer 1 Report
Comments and Suggestions for Authors
This work focuses on the real-time requirements for the query and retrieval of geohazard monitoring data and conducts research on a spatiotemporal index model called the "BCHR model." By employing the CluStream algorithm, it introduces the notions of dynamic clustering into the dynamic construction of the HRTree, and this is quite innovative. However, it has the following unclarities/shortcomings:
1. The CluStream algorithm generates clustering using K-mean, and BCHR generates leaf nodes of the HRTree based on the clustering. The issue is, for clustering based on spatial locations, how do we determine the value of K? For different types of geohazards, areas of occurrences, etc., how should K be determined? This the authors did not explain, while it is very important, as it directly determines the practical value of this work.
2. The authors only presented in figure 5 the advantages of using BCHR for generating leaf nodes of the HRTree based on clustering, as compared to directly generating leaf nodes based on the Hilbert curve construction method. However, they did not analyze the employment of BCHR from a theoretical perspective. This is also very important, as it determines the academic value of this work.
3. Since the focus of this work is on practical applications, it is necessary to analyze the specific circumstances and needs of real-world applications. This work lacks analysis in this aspect, which makes this study in want of sufficient explanations for its basis.
4. The introduction of simulated data is not very clear. For example, the author mentioned simulating 10,000 data streams, but did not elaborate on the specific characteristics of each data stream. This makes figure 7 lack a reference benchmark. In addition, the following aspects lack clarity:
(a) The simulation process and algorithm.
(b) Comparative analysis with real data.
(c) Which K-value was chosen when using CluStream.
Author Response
Dear Reviewer, we sincerely appreciate your valuable comments and suggestions. Please refer to the attached document where we have duly responded to your comments.

Reviewer 2 Report
Comments and Suggestions for Authors
I suggest correcting the brought pictures in such a way that they all use the same font. After the aforementioned recommendation, I suggest that the paper be accepted for publication.
Author Response

(The authors gave the same response as above.)

Reviewer 3 Report
Comments and Suggestions for Authors
Based on the geographical disaster monitoring data, this paper proposes a spatio-temporal index method BCHR Tree based on B+ tree and Hilbert R-tree, which is coupled with spatial sulfur clustering method, aiming at the problem that the current spatio-temporal index method is difficult to deal with real-time data and cannot perform time index. It improves the efficiency of geographic disaster monitoring data index retrieval.
1. The reference for Hilbert R-tree mentioned in the second paragraph of the manuscript introduction should be Kamel I, Faloutsos C. Hilbert R-Tree: An improved R-Tree using fractals[R]. 1993. Instead of the 19th reference in the text (R*-Tree), and there is no reference to Hilbert R-tree in the reference, which is not a good phenomenon.
2. In the third paragraph of the introduction, the R-tree index structure proposed in reference 20 is to solve the problem of spatial rectangular overlapping area of R-tree, rather than Hilbert R-tree, and the expression needs to be more rigorous. In addition, the early Chinese references in the relevant review part of R-tree are relatively large, which is difficult to reflect the current research status and cannot explain the necessity of the method to solve the problem. Moreover, only literatures that combined R-tree with clustering methods were reviewed, which lacked breadth and depth.
3. At present, there have been many articles on spatio-temporal indexing. It is suggested to supplement the similarities and differences between the proposed method and other spatio-temporal indexing methods, as well as the rationality and advantages of using B+ tree combined with Hilbert R-tree.
4. The Model Overview part of the second section of the manuscript is not very clear. Please explain the difference and relationship between the Client here and the Master and Slave in distributed computing, or is it just the terminal input of a user? Does the index affect the nodes in Hadoop? Please make it clear.
5. In section 3.2.1 of the manuscript, the micro-clustering steps used in the index are introduced, where the K-means algorithm is used. How to determine the initial value of k is not explained, whether this is an empirical parameter, or an automated value selection strategy needs to be clarified.
6. In section 3.2.3 of the manuscript, how should the index of historical data at different times be managed in the process of generating BCHR tree? Do you need to merge? If no merging is required, how do we handle queries that are at time intervals? If so, how should the two indexes be merged?
7. It is suggested to add a detailed description of the data in Subsection 4.1 of the manuscript, including the spatial distribution of the data and the size of the data volume, which are essential for judging the efficiency of the method. In addition, we know that the performance of R-trees varies greatly depending on whether the data is evenly distributed or not, so we can briefly discuss whether we should evaluate the performance of our methods when the data is not evenly distributed.
8. The method comparison part is relatively simple, only the spatial index square method is compared. It is suggested to increase the time comparison of index construction. Considering that the application scenario of the method is index construction and data storage of real-time data, it should have a greater time advantage in index construction. In addition, whether the additional storage burden caused by indexing should be considered and evaluated is recommended for analysis. Methods may need to be evaluated for efficiency and space cost when applied.
9. In the comparison of index insertion performance in Section 4.2.3 of the manuscript, the experimental Settings and parameters are not clearly explained. It is suggested to describe briefly the initial amount of data inserted by the index and the time distribution of the inserted data.
Comments on the Quality of English LanguageOverall, the quality of English expression is well. Although there is confusion in the presentation of the manuscript. MBR appears many times in the manuscript. Whether the same BCHR Tree and BCHR index are the same concept, the full name does not seem to be reflected in the text, and the abbreviation may be misleading to the reader.
Author Response

(The authors gave the same response as above.)

Reviewer 4 Report
Comments and Suggestions for Authors
The manuscript proposes a spatio-temporal indexing model called BCHR tree based on the data stream clustering algorithm CluStream for monitoring and early warning of complex geological disasters in Yunnan Province. The overall idea is reasonable and innovative. However, some issues need to be addressed:
1. The experiments are limited, only comparing with Hilbert R-tree and R-tree. It is suggested to include a comparison with other stream processing and indexing algorithms to better demonstrate the advantages of the proposed model.
2. The algorithm complexity analysis is not thorough enough. Quantitative analysis such as node split times and query time complexity should be provided.
3. The discussion on fault tolerance and scalability of the model is insufficient, which is crucial for a real-time processing system.
4. The data processing flowchart could be improved to illustrate the interactions between components more clearly.
5. Application examples and scenarios are not rich enough. It is advisable to supplement more real-world geological disaster cases to enhance applicability.
6. The conclusion is relatively simple. Further summarize the innovations, strengths, and potential improvements of the model.
7. Some key technical points are not described clearly and in detail.
Overall, the research is significant and valuable for monitoring complex geological disasters in Yunnan Province. However, more work needs to be done in terms of experimental design, algorithm analysis, applicability, etc. Please consider revising according to the comments.
Comments on the Quality of English LanguageMinor editing of English language required.
Author Response

(The authors gave the same response as above.)

Round 2
Reviewer 1 Report
Comments and Suggestions for Authors
No questions.
Reviewer 3 Report
Comments and Suggestions for Authors
The revised manuscript has made great progress. It is recommended to check the whole paper sentence by sentence.